# Template-free Articulated Neural Point Clouds for Reposable View Synthesis

Lukas Uzolas    Elmar Eisemann    Petr Kellnhofer
Delft University of Technology
The Netherlands
{l.uzolas, e.eisemann, p.kellnhofer}@tudelft.nl

## Abstract

Dynamic Neural Radiance Fields (NeRFs) achieve remarkable visual quality when synthesizing novel views of time-evolving 3D scenes. However, the common reliance on backward deformation fields makes reanimation of the captured object poses challenging. Moreover, the state of the art dynamic models are often limited by low visual fidelity, long reconstruction time or specificity to narrow application domains. In this paper, we present a novel method utilizing a point-based representation and Linear Blend Skinning (LBS) to jointly learn a Dynamic NeRF and an associated skeletal model from even sparse multi-view video. Our forward-warping approach achieves state-of-the-art visual fidelity when synthesizing novel views and poses while significantly reducing the necessary learning time when compared to existing work. We demonstrate the versatility of our representation on a variety of articulated objects from common datasets and obtain reposable 3D reconstructions without the need of object-specific skeletal templates. The project website can be found at https://lukas.uzolas.com/Articulated-Point-NeRF/.

## 1   Introduction

Synthesizing novel photo-realistic views of captured 3D scenes is important for many domains including virtual/augmented reality, video games or movie productions. In recent years, Neural Radiance Fields (NeRFs) [1] have proved their remarkable capacity to represent complex view-dependent effects, captured in photographs and videos, sparsely sampled from natural light fields [2]. Follow-up works have extended the scope to dynamic scenes [3–10], facilitating rendering of unseen views at different timestamps. Despite progress in reconstruction quality and speed [11], manipulating learned scenes remains a challenge, but would be a highly desirable feature, as it can enable downstream applications, such as the creation of avatars for virtual presence or 3D assets for games and movies.

The key challenge of reposing a NeRF is inverting the backward-warping function that maps individual observations to a shared canonical representation [3]. It is an inversion of traditional kinematic animation, such as Linear Blend Skinning (LBS) [12], where a canonical shape is forward-warped to a desired pose. Such inverse mapping often requires resolving ambiguous situations as it is difficult to guarantee bijectivity (see Fig. 2). A common remedy are parametric templates, typically built for narrow application domains, such as human heads and bodies [13–16], but they are difficult to generalize. Alternatively, object shapes can be retrieved from videos as ensembles of geometric parts, yet existing techniques provide limited image-synthesis fidelity [17–19]. Our work aims to combine these different lines of work and enable joint learning of the NeRF representation and its pose parameterization from sparse or dense multi-view videos. We aim for time-efficient class-agnostic view synthesis of reposable models with high image-synthesis quality without access to a template or pose annotation, which is a combination not currently covered by existing work (see Table 1).

37th Conference on Neural Information Processing Systems (NeurIPS 2023).

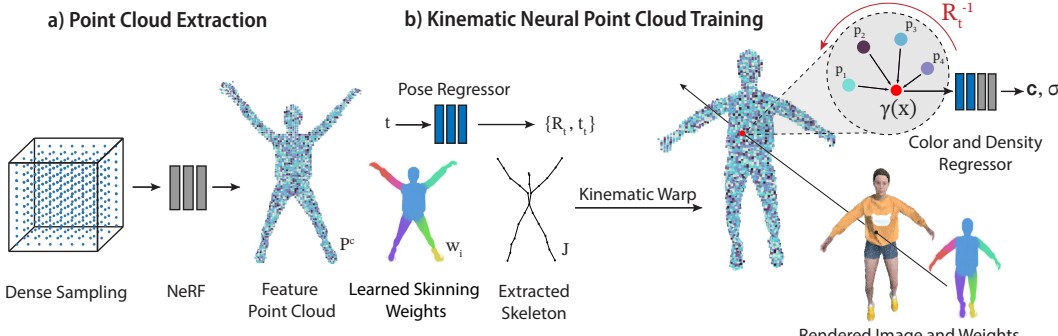

**a) Point Cloud Extraction**

**b) Kinematic Neural Point Cloud Training**

Pose Regressor

$t \longrightarrow$ ▮▮▮ $\longrightarrow \{R_t, t_i\}$

$R_t^{-1}$

$\gamma(x) \longrightarrow$ ▮▮▮ $\longrightarrow$ **c**, $\sigma$

Color and Density Regressor

Dense Sampling    NeRF    Feature Point Cloud    Learned Skinning Weights    Extracted Skeleton

Kinematic Warp

$P^c$    $w_i$    $J$

Rendered Image and Weights

Figure 1: Overview of our method: a) First, we pre-train a NeRF backbone to initialize a feature point cloud $P^c$ for a selected canonical timestamp and to extract an initial skeleton. b) During the main training stage, $P^c$ is forward-warped using LBS consisting of learned time-invariant skinning weights $\hat{w}_i$ and time-dependent pose transformations from an MLP regressor $\Phi_r$. The image is obtained by integration and decoding of features aggregated from points along each camera ray. In summary, we fine-tune the initial neural point features $\mathbf{f_i}$, skinning weights $\hat{w}_i$, joints $J$, density and color regressor $\Phi_d$ and $\Phi_c$ of the backbone. We fully train the pose regressor $\Phi_r$ and feature point decoder $\Phi_p$ from scratch. In test time, we modify the pose transformations to obtain novel poses.

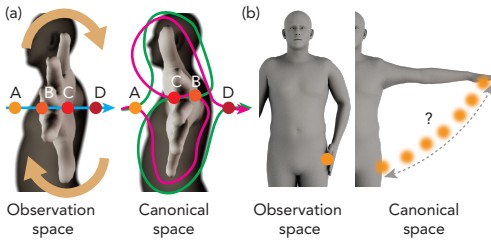

(a)    (b)

A   B C D    A   C B   D

Observation space    Canonical space    Observation space    Canonical space

Figure 2: Examples of poses difficult for backward warping. (a) Ill-defined projection from an observation to the canonical space. Both ambiguous solutions (green and magenta) correctly pass through the semantically corresponding surface points B and C. (b) Projection ambiguity for points of contact between two surfaces. Note that in contrast, it is trivial to forward warp the object points from a well-chosen canonical space to any observation space.

| Method | Pose | Generic | Fildelity | Training |
|---|---|---|---|---|
| **LASR** [17] | No | Yes | Shape | $\leq 2\,\text{h}$ |
| **ViSER** [18] | No | Yes | Shape | $2\text{–}12\,\text{h}$ |
| **BANMO** [22] | No | Yes | Low | $\geq 12\,\text{h}$ |
| **D-NeRF** [3] | No | Yes | High | $\geq 12\,\text{h}$ |
| **TiNeuVox** [11] | No | Yes | High | $\leq 2\,\text{h}$ |
| **CASA** [23] | Yes | No | Shape | $\leq 2\,\text{h}$ |
| **LASSIE** [19] | Yes | Yes | Low | $\geq 12\,\text{h}$ |
| **WIM** [20] | Yes | Yes | High | $\geq 12\,\text{h}$ |
| **Ours** | Yes | Yes | High | $\leq 2\,\text{h}$ |

Table 1: A comparison of representative dynamic 3D representations learned from 2D videos. We analyze reposibility, agnosticism to object class, image synthesis fidelity and training time. Papers that demonstrate reposing are shown below the bar. "*Shape*" denotes shape-only reconstruction without texture details.

Recent work by Noguchi et al. [20] addresses a similar problem. However, we exploit the structure-free nature of point-based NeRF representations [21], enabling direct forward warping of the canonical object to any desired pose, while maintaining the capacity and flexibility of NeRF-like rendering to capture highly detailed image features. Together with our automatically extracted and jointly-optimized LBS skeletal pose parameterization, our work allows for faster training and achieves state-of-the-art visual quality in a much shorter time. Finally, we demonstrate how to easily repose learned representations of varied objects by directly editing joint angles of the forward LBS model.

To summarize, we make the following contributions: 1) We propose a novel method for learning articulated neural representations of dynamic objects using forward projection of neural point clouds. 2) We demonstrate state-of-the-art novel view synthesis for a variety of multi-view videos with training times lower than comparable methods. 3) We jointly learn a skeletal model without domain-specific priors or user annotations and demonstrate reposing of the reconstructed 3D objects.

## 2 Related Work

**Neural Radiance Fields and Parameterizations**   Implicit neural networks have recently emerged as a powerful tool for building differentiable representations of 3D scenes [24–47] and for learning view-dependent *Neural Radiance Fields* (NeRFs) [1, 2, 48–57] enabling photo-realistic novel view synthesis. Their high computational cost has been dramatically reduced by spatial decomposition [58–60] and hybrid representations [61–65].Recently, neural point clouds have combined expressive neural functions with the flexibility of explicit geometry [21, 66–69]. We also use neural point clouds, but unlike previous work, we jointly learn a forward kinematic model along with the associated soft part segmentation and, thus, convert fast voxel-based representations to reposable NeRFs.

**Dynamic Neural Radiance Fields**   Equivalent to 2D video, dynamic NeRFs capture temporal changes in scenes by encoding each frame separately [70], expanding the radiance field into the temporal domain [7, 8, 71, 72], or time-dependent interpolation in a compact latent representation [6, 9, 10]. Alternatively, a single canonical NeRF can be animated by backward warping from the canonical space to individual time-varying observation spaces [3–6, 9, 11]. However, such warps rely on the bijectivity of the mapping which is difficult to guarantee for all observed poses (see Fig. 2). Forward mapping only requires a single well-posed canonical representation, which we exploit.

**Object Reposing**   Directly reposing high-dimensional deformation fields of dynamic NeRFs is impractical. Instead, parametric templates can sparsely represent prominent deformation modes for faces [13, 14], bodies [15, 16], hands [73], and also non-human objects, such as animals [74]. Together with skeleton-based LBS, they enable articulated neural representations of human heads [69, 75–81] or bodies [82–97], as well as modeling distributions in generative models [98–101]. However, the assumption of piece-wise rigid motion and the availability of a skeletal template restrict the applicable object classes. The former issue has been addressed by physically inspired but computationally expensive deformation models [102]. To remedy the latter issue, the skeletal model can be retrieved from a database [23], adapted from a generic graph [19, 103], fitted as a template to 2D observations [104–108] or fully learned during a 3D shape recovery [20, 109]. An external cage can also be used for re-animation of static NeRFs [110]. Alternatively, a surface embedding can be learned without a skeleton if reposing is not required [17, 18, 22].

Similarly to Noguchi et al. [20], we jointly learn a 3D object representation, a skeletal model, and observed poses. However, we utilize a point-based NeRF representation to benefit from its computational efficiency, ease of geometric transformation, and capacity to learn complex light fields.

We share our point-based approach with a contemporaneous work *MovingParts* [111] which, however, relies on inverting the backward flow. We demonstrate the reconstruction of large transformations in the *Robots* dataset [20], not demonstrated in *MovingParts*. Furthermore, our approach does not enforce binary segmentation and, hence, can represent non-rigid part transitions (see Fig. 6 right).

## 3 Preliminaries

Our method builds upon available NeRF reconstruction methods as a backbone for learning a reposable dynamic representation. In principle, any NeRF-like representation is a suitable initial point for our training. In practice, we use TiNeuVox [11] in all experiments, as it enables fast reconstruction with both sparse and dense multi-view supervision in dynamic scenes. Next, we describe the core concepts of NeRF [1] and TiNeuVox [11] to provide context for our method in Sec. 4.

**Time-Aware Neural Voxels**   NeRF [1] maps a point $\mathbf{p} = (x, y, z)$ and view direction $\mathbf{d} = (\theta, \phi)$ to color $\mathbf{c} = (r, g, b)$ and density $\sigma$. The mapping function often takes form of a Multi-Layer-Perceptron (MLP) $\Phi$, such that $(\mathbf{c}, \sigma) = \Phi(\mathbf{p}, \mathbf{d})$. The color of an image pixel $C(\mathbf{r})$ can be obtained via volume rendering along the ray $r(t) = \mathbf{o} + t\mathbf{d}$.

TiNeuVox [11] expands NeRF to dynamic scenes and significantly increases training speed by utilizing an explicit volume representation. To this extent, each point $\mathbf{p}$ at time $t$ in observation space is backward-warped to canonical space as

$$\mathbf{p}^c = \Phi_b(\hat{\mathbf{p}}, \hat{\mathbf{t}}), \text{ where } \hat{\mathbf{p}} = \gamma(\mathbf{p}), \ \hat{\mathbf{t}} = \Phi_t(\gamma(t)), \tag{1}$$

Here, $\gamma$ is the positional encoding [1], $\Phi_t$ a small color-decoding MLP, $\Phi_b$ a backward-warping MLP, and $\hat{\mathbf{p}}$ and $\hat{\mathbf{t}}$ the encoded point and time respectively. The canonical point $\mathbf{p}^c$ is used to sample a feature vector $\mathbf{v}_m \in \mathcal{R}^V$ from the explicit feature volume $\mathbf{V} \in \mathcal{R}^{X \times Y \times Z \times V}$ by means of trilinear interpolation $interp$ with varying stride $s_m$ at each scale: $\mathbf{v}_m = \gamma(interp(\mathbf{p}^c, \mathbf{V}, s_m))$. The feature vectors across all scales are concatenated as $\mathbf{v} = \mathbf{v}_1 \oplus ... \oplus \mathbf{v}_m \oplus ... \oplus \mathbf{v}_M$, where $\oplus$ is the concatenation operator. Finally, the view-dependent color and density are regressed by additional MLPs $\Phi_f$, $\Phi_d$ and $\Phi_c$:

$$\mathbf{f} = \Phi_f(\mathbf{v}, \hat{\mathbf{p}}, \hat{\mathbf{t}}), \text{ where } \sigma = \Phi_d(\mathbf{f}), \ \mathbf{c} = \Phi_c(\mathbf{f}, \mathbf{d}). \tag{2}$$

The volume rendering equation [112] integrates density/color of points $\mathbf{p}_i$ sampled along the ray $r$:

$$\hat{C}(\mathbf{r}) = \sum_{i=1}^{N} T_i(1 - exp(-\sigma_i \delta_i))\mathbf{c}_i, \text{ where } T_i = exp(-\sum_{j=1}^{i-1} \sigma_j \delta_j)), \tag{3}$$

Here, $\delta_i$ is the distance between $\mathbf{p}_i$ and $\mathbf{p}_{i+1}$. The feature volume $\mathbf{V}$ and all MLPs are optimized end-to-end and supervised by the Mean Squared Error (MSE) and the corresponding ground-truth pixel color $C(\mathbf{r})$: $\mathcal{L}_{photo} = ||\hat{C}(\mathbf{r}) - C(\mathbf{r})||_2^2$.

Note that because $\Phi_f$ (Eq. 2) is conditioned by the time $\hat{\mathbf{t}}$, space and time are entangled and the feature volume $V$ does not represent a single static canonical shape. Therefore, we cannot directly learn a forward warping function to invert $\Phi_b$ as proposed by Chen et al. [102]. In the next section, we show that this is not necessary and we can learn a forward kinematic model directly without relying on specific properties of the backbone, which will allow us to replace it in the future.

## 4 Method

While dynamic NeRF models such as TiNeuVox [11] can reproduce motion in a dynamic scene, they do not enable reposing. Furthermore, it is impractical to post hoc reparametrize their motion representation due to the ambiguities of backward flow inversion (see Fig. 2).

We instead propose a completely new representation based on a neural point cloud and an explicit kinematic model for forward warping, and we use the backbone only as an initialization for our training procedure (see Fig. 1). First, we extract a feature point cloud from a selected canonical frame of the backbone Sec. 4.1. Second, we describe the underlying kinematic model of our 3D representation and its initialization Sec. 4.2. Consecutively, we introduce how the 3D point cloud is warped from a canonical space to an observation space and rendered Sec. 4.3. Lastly, we describe our losses Sec. 4.4.

### 4.1 Canonical Feature Point Cloud

We first pre-train a backbone NeRF model (TiNeuVox [11]) using its default parameters. To initialize the feature point cloud, we sample the canonical density function of TiNeuVox $\sigma = \Phi_d(\mathbf{f})$ (Eq. 2) on a uniform coordinate grid and discard empty samples through thresholding. The grid resolution is adaptively chosen, such that $|P| \approx 10000$, similarly to other explicit NeRFs [11, 64, 113]. After thresholding, we obtain points $P = \{\mathbf{p}_i | i = 1, ..., N\}$. Furthermore, we extract a feature vector $\mathbf{f}_i$ for each point $\mathbf{p}_i$ (see Eq. 2).

### 4.2 Kinematic Model

We forward-warp our feature point cloud from canonical space to observation space using an LBS kinematic model to match the training images. Here, we describe how we initialize, use, and simplify our kinematic model.

**Skeleton Initialization** We do not rely on a class-specific template to initialize our kinematic skeleton. Instead, we extract an approximate initial skeleton tree by Medial Axis Transform (MAT) and refine it during training.

Let $M = \{\mathbf{p}_m | m = 1, ..., M\}$ be the set of medial axis points extracted by applying a MAT on $P^c$. We choose the root of our kinematic model $\mathbf{p}_{root}$ as the medial axis point that is closest to all other medial-axis points, i.e., $\mathbf{p}_{root} := \arg\min_i \sum_j ||\mathbf{p}_i - \mathbf{p}_j||_2, \forall \mathbf{p}_i \in M, i \neq j$.

Next, we leverage dense sampling-grid neighborhoods and define a graph $G_M$ connecting neighboring points in $M$. Points disconnected from $\mathbf{p}_{root}$ are removed. We then use a heuristic to select sparse joints (nodes) as a subset of $G_M$ and define the bones (edges) based on their connectivity. To this extent, we traverse $G_M$ starting from $\mathbf{p}_{root}$ in a breadth-first manner and mark $\mathbf{p}_m$ as a joint $\mathbf{j}_b$ if its geodesic distance from the preceding joint exceeds a threshold $B_{length} = 10$. As a result, we obtain a set of time-invariant canonical joint positions $J = \{\mathbf{j}_b\}$ which we further optimize during training.

While this skeleton is usually over-segmented, we show in our experiments that this does not hamper the training. We propose a pruning strategy to enable easier pose manipulation afterwards (see Fig. 7).

**Blend Skinning Weights** For each point $\mathbf{p}_i$ we initialize its raw skinning weight vector $\hat{\mathbf{w}}_i$ by an exponential decay function of the distance $dist$ to each bone line $b_j$ such that $\hat{w}_{i,j} = 1/e^{dist(\mathbf{p}_i^c, b_j)}$. Before skinning, we obtain the final blend skinning weight vector $\mathbf{w}_i$ through scaling by a global learnable parameter $\alpha$ and applying a softmax: $w_{i,j} = e^{\hat{w}_{i,j}/\alpha}/\sum_k e^{\hat{w}_{i,k}/\alpha}$. During the training, we optimize the initial $\hat{\mathbf{w}}_i$ as well as $\alpha$. Accounting for the per-point weights, we define our full canonical feature point cloud as $P^c = \{(\mathbf{p}_i^c, \mathbf{f}_i, \hat{\mathbf{w}}_i) | i = 1, ..., N\}$.

**Point Warping** We forward-warp the canonical point cloud $P^c$ to an observation space of timestamp $t$ via LBS [12]. The local transformation matrix $\hat{T}_b^t$ of each bone $b$ is defined by a rotation $R_b^t$ around its parent joint $\mathbf{j}_b$. Consequently, each point $\mathbf{p}_i^c$ is transformed by a linear combination of bone transformations as:

$$\mathbf{p}_i^t = \bar{T}_i^t p_i^c = \sum_{b=1}^{|B|} w_{i,b} T_b^t \mathbf{p}_i^c, \text{ with } T_b^t = T_{p_b}^t \hat{T}_b^t \text{ and } \hat{T}_b^t = \begin{bmatrix} R_b^t & \mathbf{j}_b + R_b^t \mathbf{j}_b^{-1} \\ \mathbf{0} & 1 \end{bmatrix}, \tag{4}$$

where $T_b^t$ is defined recursively by its parent bone $p_b$. $T_{p_b}^t$ of the skeleton root is identity.

We express rotations $R_b^t \in SO(3)$ using the Rodrigues' formula, where $\hat{\mathbf{r}} = \mathbf{r}/||\mathbf{r}||$ is the axis of rotation. However, we learn the rotation angle $\theta$ directly as an additional parameters because we find it leads to a better pose regression than using $\theta = ||\mathbf{r}||$. The time-dependent rotations $\mathbf{r}_b, \theta_b$ for each bone $b$ are regressed by an MLP: $\Phi_r(\gamma(t)) = \mathbf{r}_1^t, \theta_1^t, ..., \mathbf{r}_b^t, \theta_b^t, ..., \mathbf{r}_B^t, \theta_B, \mathbf{r}_r^t, \theta_r^t, \mathbf{t}_r^t$, where $\mathbf{r}_r^t$, $\theta_r^t$ and $\mathbf{t}_r^t$ are the time-dependent root rotation and translation.

**Skeleton simplification** The initial skeleton's over-segmentation does not hamper pose reconstruction during training, but we optionally simplify the skeleton after training to ease pose editing (see Fig. 7). We prune or merge bones based on the size of rotation angles $\theta_b^t$ produced by the transformation regressor $\Phi_r$. Joints, which do not exhibit a rotational change from the rest pose above the threshold of $t_r$ deg in more than 5% of the observed timestamps, are marked static. We then merge the bones of such joints and their corresponding weights $\hat{\mathbf{w}}$. See the supplement for details.

### 4.3 Dynamic Point Rendering

We adopt the point cloud rendering from [21] but extend it to explicitly model rotational invariance of the radiance field. For each sampling point $x$, we find up to $N = 8$ nearest observation feature points $\mathbf{p}_i^t$ within a radius of $0.01$ and roto-translate them into their canonical frames. This enables the feature embedding MLP $\Phi_p$ to learn spatial relations in a pose-invariant coordinate frame as:

$$\mathbf{f}_{i,x}^t = \Phi_p(\mathbf{f}_i, x_{\mathbf{p}_i}^t), \text{ with } x_{\mathbf{p}_i}^t = \gamma(R_i^{t^{-1}}(x - \mathbf{p}_i^t)), \tag{5}$$

where, $R_i^t$ is the rotation component $\bar{T}_i^t$ (Eq. 4) for point $\mathbf{p}_i^t$ and $\gamma(.)$ is the positional encoding [1].

The neighboring embeddings $\mathbf{f}_{i,x}^t$ are aggregated by inverse distance weighting, which produces the final feature input for $\Phi_d$ and $\Phi_c$ (see Eq. 2) and the consequent volume rendering (Eq. 3):

$$\mathbf{f}_x^t = \sum_i^N \frac{d_i}{\sum_j^{|B|} d_j} f_{i,x}^t, \text{with } d_i = ||p_i^t - x||^{-1}; \sigma = \Phi_d(\mathbf{f}_x^t); \mathbf{c} = \Phi_c(\mathbf{f}_x^t, \mathbf{d}). \tag{6}$$

### 4.4 Losses

Next to the photometric loss $\mathcal{L}_{photo}$ (Sec. 3), we utilize a 2D chamfer loss to penalize the difference between the point cloud projected into a training view $I(P^t)$ and the corresponding 2D ground truth

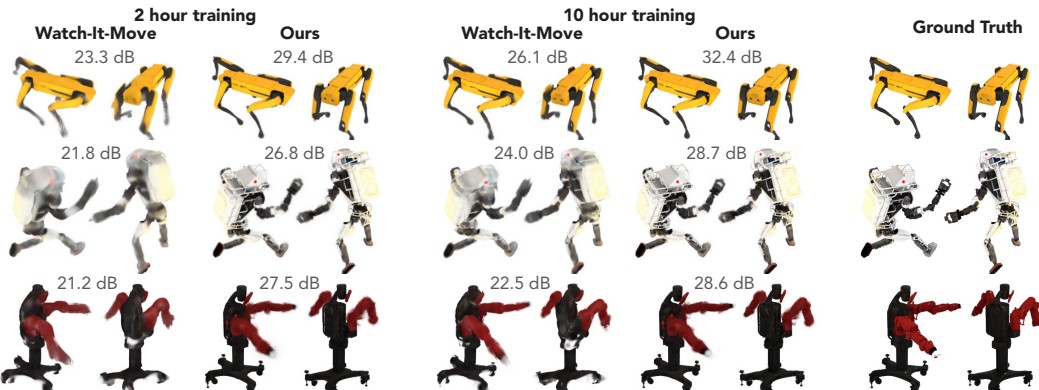

Figure 3: Qualitative comparison displaying two held-out views-frames of scenes from the *Robots* rendered by WIM [20] and our method after 2 and 10 hours of training, and the PSNR scores.

object mask $M^t$: $\mathcal{L}_{mask} := \mathcal{L}_{chamf}(I(P^t), M^t)$. The chamfer loss is defined as:

$$\mathcal{L}_{chamf}(P^1, P^2) = \frac{1}{|P^1|} \sum_i^{|P^1|} \min_j ||\mathbf{p}_i^1 - \mathbf{p}_j^2||_2^2 + \frac{1}{|P^2|} \sum_j^{|P^2|} \min_i ||\mathbf{p}_i^1 - \mathbf{p}_j^2||_2^2. \tag{7}$$

To prevent the skeleton from diverging too much from the medial axis $M$, we further minimize the chamfer loss between $M$ and the joints $J$ (see Sec. 4.2): $\mathcal{L}_{skel} := \mathcal{L}_{chamf}(M, J)$. In addition, we minimize the transformation angles and the root translation: $\mathcal{L}_{tranf} = (\sum_i |\theta_i^t|) + |\mathbf{t}_r^t|$. Local rigidity is further enforced upon points after deformation via as-rigid-as-possible regularization:

$$\mathcal{L}_{arap} = \sum_i^{|P^t|} \sum_j^{N(p_i)} |||p_i^c - p_j^c||_2^2 - ||p_i^t - p_j^t||_2^2|. \tag{8}$$

Lastly, we apply two regularizers on the blend skinning weights. To encourage smoothness, we penalize divergence of skinning weights in the rendering neighborhood $N$: $\mathcal{L}_{smooth} = \sum_i^{|P^t|} \sum_{j \in N(p_i)} |w_i - w_j|$, and, to encourage sparsity, we apply:

$$\mathcal{L}_{sparse} = -\sum_i^{|P^c|} \sum_j^{B} w_{i,j} \log(w_{i,j}) + (1 - w_{i,j}) \log(1 - w_{i,j}). \tag{9}$$

In total, our training loss is $\mathcal{L} = \omega_0 \mathcal{L}_{photo} + \omega_1 \mathcal{L}_{mask} + \omega_2 \mathcal{L}_{skel} + \omega_3 \mathcal{L}_{tranf} + \omega_4 \mathcal{L}_{smooth} + \omega_5 \mathcal{L}_{sparse} + \omega_6 \mathcal{L}_{ARAP}$ where $\omega = \{200, 0.02, 1, 0.1, 10, 0.2, 0.005\}$ in our experiments.

## 5 Experiments

Here, we evaluate our work, which obtains state-of-the-art view-synthesis quality with a lower training cost than other articulated methods. We also demonstrate class-agnostic reposing capability and we evaluate contribution of method components in an ablation study. Video examples can be seen on the project website.

**Datasets**  We chose three multi-view video datasets that are commonly used for the evaluation of dynamic multi-view synthesis methods and pose articulation. First, the *Robots* dataset [20] features multi-view synthetic videos of 8 topologically varied robots, making it well suited for testing pose articulation performance (see Fig. 4). We use 18 views for training and 2 for evaluation. Second, the *Blender* dataset [3] is a sparse multi-view synthetic dataset with 5 humanoid and 2 other articulated objects[1]. Its visual quality benchmarks are used to test the fidelity of image detail reconstruction (see

---

[1]We do not use the multi-component *Bouncing balls* scene.

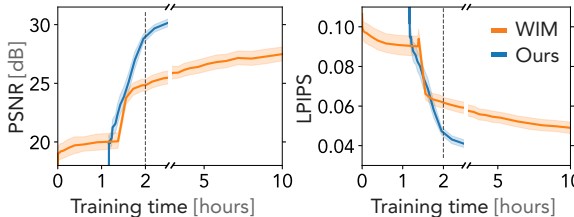

| Time | Method | PSNR↑ | SSIM↑ | LPIPS↓ |
|------|--------|-------|-------|--------|
| 2 h | WIM [20] | 25.05 | 0.952 | 0.059 |
| | Ours | **28.81** | **0.968** | **0.047** |
| 10 h | WIM [20] | 27.61 | 0.964 | 0.046 |
| | Ours | **30.19** | **0.973** | **0.041** |

Figure 4: Quality of unseen view synthesis during training with 95% confidence intervals in the *Robots* dataset [20]. The initial plateau of WIM [20] matches the 10k warm-up steps used by the authors before training with all data. Our onset time corresponds to the 70 minutes required for pre-training of the backbone. Training of our method was terminated after 2.5 hours.

| Method | Reposable | PSNR↑ | SSIM↑ | LPIPS↓ | Training Time |
|--------|-----------|-------|-------|--------|---------------|
| D-NeRF [3] | No | 30.50 | 0.95 | 0.07 | 20 hours |
| TiNeuVox-B [11] | No | 32.67 | 0.97 | 0.04 | 28 mins |
| Hexplane [72] | No | 31.04 | 0.97 | 0.04 | 11.5 mins |
| K-Plane hybrid [71] | No | 31.61 | 0.97 | - | 52min |
| WIM* [20] | **Yes** | 23.81 | 0.91 | 0.10 | 11 hours |
| Ours* | **Yes** | 29.10 | 0.94 | 0.06 | 2.5 hours |

Table 2: Quality of unseen view synthesis for the *Blender* dataset [3]. Results of D-NeRF and TiNeuVox-B reprinted from Fang et al. [11]. *Without the "Bouncing balls" scene.

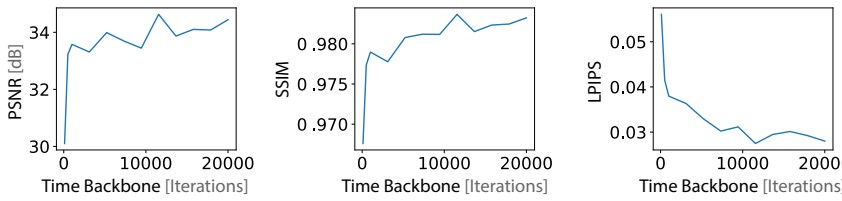

Figure 5: Effect of the backbone initialization pre-training steps on the final result of our method when trained on the *Jumping Jacks* scene from the *Blender* dataset. Our final choice of 20k iterations corresponds to approximately 70 minutes of real time.

Fig. 6 right). We use the original training-test split. Third, *ZJU-MoCap* dataset [82] is a common test suite for dynamic human reconstruction and we evaluate 5 of its multi-view sequences with the same 6 training views as used in *Watch-It-Move* [20]. Finally, we evaluate image quality using peak signal-to-noise ratio (PSNR), structural similarity (SSIM) [114], and learned perceptual image patch similarity (LPIPS) [115] image metrics.

**Implementation details** We pre-train the TiNeuVox [11] backbone using the authors' implementation and an additional distortion loss [116], as implemented in [117]. Our method is implemented in Pytorch and we train each scene using the Adam optimizer for 160k (*Blender* and *Robots*) or 320k (*ZJU-MoCap*) iterations with a batch size of 8192 rays, sampled randomly from multiple views and a single timestamp. We choose the canonical timestamp by visual inspection, and gradually increase the number of observed timestamps during training. We adjust ray sampling and scheduling for the *ZJU-MoCap* dataset (see the Supplement). All experiments were done on a single Nvidia GPU RTX 3090Ti. See the supplementary materials for details. For more details, see the Appendix.

**Baselines** We compare our method to state-of-the-art non-articulated and articulated methods for high-fidelity multi-view video synthesis (see Table 1 for an overview). *D-NeRF* [3] extends NeRF to the temporal domain by backward warping a static canonical NeRF. *TiNeuVox* [11] improves performance of *D-NeRF* using voxel grids. *Hexplane* [72] and *K-Planes* [71] decompose to the space-time volume to several hyper-planes. Finally, *Watch-It-Move* [20] (WIM) jointly learns a surface representation and LBS model for articulation. Note, that because we aim at time-efficient

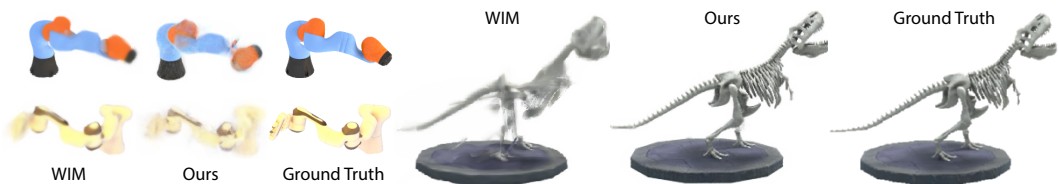

WIM                    Ours                    Ground Truth

WIM          Ours          Ground Truth

Figure 6: While our model well reproduces details and non-rigidity of the *Dinosaur* (right), it can fail for complex motions combining long chains of rotations with texture cue ambiguity (left).

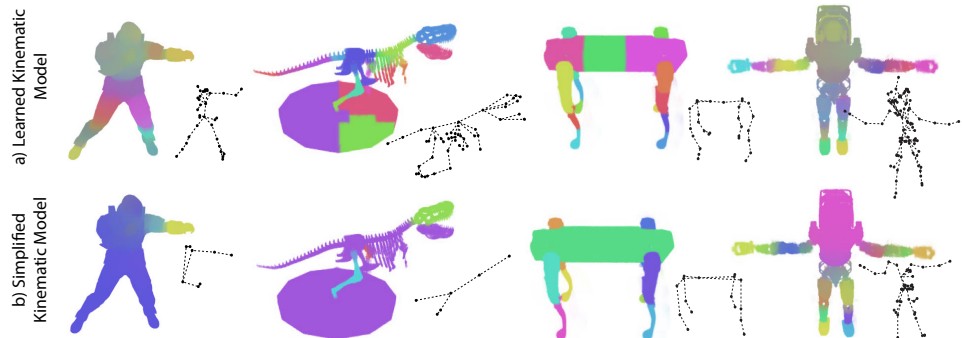

a) Learned Kinematic Model

b) Simplified Kinematic Model

Figure 7: Learned LBS weights and skeleton: a) After training. b) After additional post-processing (weight merging and skeleton pruning, see Sec. 4.2).

learning, training of WIM was stopped after 11 hours (i.e., 80k of the original 400k optimization steps).

**Novel view synthesis**  We provide results for the *Robots* view synthesis without the skeleton simplification; quantitatively in Fig. 4 and qualitatively in Fig. 3. More visual examples are available in the supplement. After the initial pre-training, our method quickly surpasses the image quality of WIM [20]. The mean image scores obtained for our method after 2 hours of training (incl. pre-training) are higher than those that WIM achieves after 10 hours. We attribute this to NeRF's capability to approximate even complex shapes, which are difficult to fit using the signed distance function utilized by WIM. Nevertheless, for visually simple objects with long nested pose transformation chains, this capacity might encourage false explanations of the articulation and cause artifacts, as visible in Fig. 6 left. However, this is not a common issue. We illustrate it in the *Blender*, where WIM struggles to represent fine details (see Fig. 6 right), while our method achieves image scores close to those of non-reposable baselines (see Table 2 and the supplement).

Furthermore, Fig. 5 shows that our image quality is not highly dependent on the pre-training phase. We observe that mere 100 iterations of *TiNeuVox* pre-training provide an initialization sufficient for achieving PSNR scores over 30 dB by our method. However, while fine geometric details are still recovered, the coarse initial shape limits the skeleton complexity and, therefore, we opt for 20k pre-training steps in all other experiments.

**ZJU-MoCap**  In Fig. 10, we compare our method to WIM in the *ZJU-MoCap* dataset. We observe that both methods are able to recover the 3D shape and the skeletal motion despite the difficulty of an accurate fine texture detail reconstruction. This can partly be attributed to imperfections in camera calibrations (see Supplement F of [96]) and partly by soft deformations of clothes which are modeled neither method. Notably, with a small modification our method learns to partially compensate for this. In Ours$^{pose}$, we condition the feature embedding network $\Phi_p$ by the skeletal poses jointly learned from scratch by our method. This improves the image quality by modeling the residual deformations. See the Supplement for details. Finally, Ours$^{SMPL}$ shows that replacing our skeleton in Ours$^{pose}$ with a ground truth from an annotated SMPL template [15] does not affect the performance. This validates our skeleton initialization based on Medial Axis Transform.

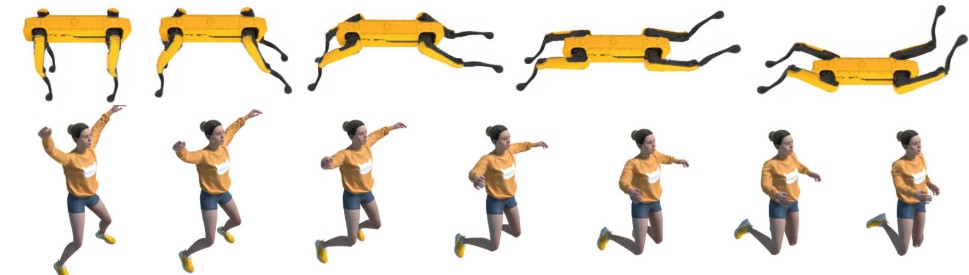

Figure 8: Reposing using the simplified skeleton. Interpolation from canonical to novel pose.

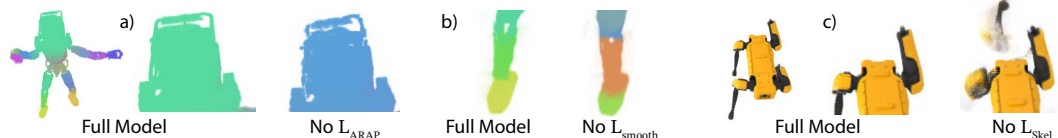

Full Model    No $L_{ARAP}$    Full Model    No $L_{smooth}$    Full Model    No $L_{Skel}$

Figure 9: Ablation examples. a) $\mathcal{L}_{ARAP}$ enforces rigidity after pruning (see upper pipes), b) $\mathcal{L}_{smooth}$ results in better part-segmentation, c) $\mathcal{L}_{skel}$ enforces the joints to stay inside the shape.

| Method | PSNR↑ | SSIM↑ | LPIPS↓ |
|---|---|---|---|
| WIM [20] | 31.08 | 0.963 | **0.053** |
| Ours | 29.60 | 0.958 | 0.063 |
| Ours$^{pose}$ | **32.05** | **0.967** | 0.056 |
| Ours$^{SMPL}$ | 32.01 | 0.967 | 0.056 |

Figure 10: Comparison in the *ZJU-MoCap* dataset. Ours: Our full method. Ours$^{pose}$: Ours with an additional pose-conditioned feature embedding network $\Phi_p$. Ours$^{SMPL}$: Ours$^{pose}$ with ground truth SMPL [15] skeleton.

**Reposing** In Fig. 7, we visualize the learned blend-skinning weights $\hat{\mathbf{w}}_i$ and skeletons with and without the skeleton simplification. The algorithm is able to substantially reduce the skeleton complexity, while largely preserving semantic articulations observed in the training data. However, it is not able to remove all unnecessary skeletal branches when complex geometry is present (see Fig. 7 right). In Fig. 8, we show that such post-processed skeletons allow for animating of novel poses by smoothly interpolating between user-defined poses. More animations can be found in the Supplement.

**Loss ablation** Our experiments suggest the regularization does not improve the image quality, but it improves the quality of the kinematic model, which is important for our main goal of reposing (see Fig. 9). Specifically, $\mathcal{L}_{ARAP}$ avoids non-rigid distortions (a), $\mathcal{L}_{smooth}$ leads to a more semantically meaningful part segmentation (b), $\mathcal{L}_{skel}$ encourages functional placement of the skeleton joints inside the object volume even for thin parts (c), and $\mathcal{L}_{tranf}$ leads to a reduction of necessary object parts after simplification (62 components are removed instead of 49 for *Atlas*). More details are provided in the Supplement.

**Backbone** The *TiNeuVox* backbone allows us to transfer parameters learned during the pre-training and initialize the features $\mathbf{f}_i$ and the regressors $\Phi_d$ and $\Phi_c$. Interestingly, our experiment shows that such off-the-shelf features $\mathbf{f}_i$ often do not need any further fine-tuning (see Table 4). Despite this, our method is agnostic to the backbone choice by design and the end-to-end training procedure of the entire model is essential for this. We demonstrate it by training with a random initialization. This way, only the point positions $\mathbf{p}_i^c$ obtainable by any 3D reconstruction method are needed. Table 3 shows that this leads to only a negligible drop in reconstruction quality in the *Robots* dataset. Here, the weight of $\mathcal{L}_{skel}$ ($w_2 = 1$) was adjusted to prevent drift of the skeleton.

| | PSNR↑ | SSIM↑ | LPIPS↓ |
|---|---|---|---|
| **Full Method** | 30.19 | 0.973 | 0.041 |
| **Random Init.** | 29.97 | 0.971 | 0.045 |

Table 3: Results with a *TiNeuVox* initialization and with a random initialization of the features $\mathbf{f}_i$ and the regressors $\Phi_d$ and $\Phi_c$ in the *Robots* dataset.

| | PSNR↑ | SSIM↑ | LPIPS↓ |
|---|---|---|---|
| **Full Method** | 29.10 | 0.94 | 0.06 |
| **No Fine-tuning** | 29.16 | 0.94 | 0.05 |

Table 4: Results with and without fine-tuning of the feature points $\mathbf{f}_i$ in the *Blender* dataset.

## 6 Discussion

**Limitations and Future Work**   We demonstrate fast learning of articulated NeRFs for state-of-the-art view synthesis and straightforward skeletal reposing for objects with piece-wise rigid pose transformations. While the LBS allows for fitting partially non-rigid deformations (see Fig. 6 right), representing fully non-rigid, topologically varying, and multi-component objects would benefit from a higher-dimensional parameterization. Chen et al. [102] aim in that direction, but an intuitive and cost-effective reposing still favors skeleton-based techniques such as ours.

The performance of our method depends on the quality of the backbone reconstruction. While TiNeuVox [11] supports reconstruction from even sparse data, a different backbone could offer a more robust starting geometry and improve the skeleton initialization. Although our approach works well even for highly structured shapes and thin structures (see Fig. 6 right), we observe incorrect joint placement for objects with long chains of highly nonlinear geometrical transformations (see Fig. 6 left). Moreover, our approach successfully reduces the number of extraneous bones but it is not able to completely eliminate all superfluous skeletal elements (see Fig. 7).

Our method is restricted to the kinematic motion space exhibited in the training sequences. While extrapolation is possible to some extent, the proposed method is not able to generalize to arbitrary unseen poses. Finally, we focus on image synthesis for user-defined poses rather than a direct fitting of unseen skeletal poses from images or motion capture. An inverse fitting of skeletal poses to novel 2D observations or transfer of poses from one object to another remain opportunities for future research.

**Conclusion**   We presented a method for fast learning of articulated models for high-quality novel view synthesis of captured 3D objects. Our forward-warped neural point clouds avoid the pitfalls of backward warping and elegantly integrate a skeleton-based LBS. As a result, our method merges straightforward reposing even of strongly animated inputs, as present in the *Robots* dataset, with high visual fidelity of NeRF rendering. Our work is a significant step towards low-cost acquisition of animatable 3D objects for games, movies, and education.

**Ethical Considerations**   Our method renders novel views and poses of 3D objects including human bodies. However, we do not focus on this class and we note that many human-specific methods exist (see Sec. 2). Nevertheless, we acknowledge that our method could potentially be used to produce fake images of people and that research is needed to understand the risks and their mitigation.

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
