# OpenReview forum: "Template-free Articulated Neural Point Clouds for Reposable View Synthesis"
_NeurIPS.cc/2023/Conference — NeurIPS 2023 poster_

### Official Review · Reviewer_u7Dh · 2023-06-22

**Soundness:** 3 good
**Presentation:** 4 excellent
**Contribution:** 3 good
**Rating:** 6
**Confidence:** 5

**Summary:**

The authors present a method for dynamic radiance fields of subjects whose motion can be described by skeletal animation. The method automatically extracts a skeleton using medial axis transform. Further it obtains a object feature point cloud from a pre-trained NeRF and expresses their position as a function of the skeleton via linear blend skinning.
The animatable point cloud (including skinning weights and animation) is optimized using the PointNeRF renderer using losses for RGB, rigidity, animation smoothness and 2D CD against the mask. After optimization, it can be manually reposed and rendered from novel views.

The method is evaluated on the Robots and Blender datasets where it compares favorably against previous work.



**Strengths:**

- Using a point based representation to solve the given task is a good idea and makes sense, given that deformations might be very local and not restricted to a smaller set of rigid parts.
- The ability for automatic part decomposition and reposing are interesting contributions that adds value with respect to other dynamic NeRFs that might produce better visual results.
- The method compares favorably against the most relevant previous work WIM.
- The extracted part decompositions and skinning weights look very good and intuitive
- The paper is well written and very clear.
- The authors clearly discuss limitations, show failure cases and provide ablations


**Weaknesses:**

- The whole work is purely constructive (not many novel conceptual insights) and most parts of the presented method are existing concepts: The method combines NeRF (as TiNeuVox), PointNeRF (adding rotation equivariance), MAT, and ARAP. However, there is novelty in the joint system. I think the positive aspects of this work outweigh this.
- The method in general is pretty close to some of the methods for animatable human radiance fields [80-93]. It differs mostly in that it does not rely on a human template but extracts one from scratch and thus, also works on other subjects. The paper would profit from a comparison with one of them on a human subject though.
- In general, while relevant related work is mentioned, it is not always compared against. Some relevant examples are NeRFies [6] and NSFF [4], which both are known for producing better results than D-NeRF, which is used in the comparisons.


**Questions:**

It does not become clear to me what exactly is done with the features $\mathbf{f}_i$ coming from TiNeuVox. Are they just used as point feature initialization and then further optimized using PointNeRF or are they kept fixed? I can imagine that the PointNeRF renderer could adapt to these features without modifying them. This boils down to the question of which parameters of the model are optimized when, which seems to be not completely clear in the paper.


**Limitations:**

Limitations and potential negative societal impact are discussed.

---

> ### Author Rebuttal · Authors · 2023-08-09
>
> Thank you very much for your feedback! We answer your specific questions in this document. Please refer to the shared response for the experiment results and for answers to common questions.
>
> **Human Subject Comparison**
>
> We added additional results for our method applied to camera captured human full-body sequences from the ZJU-Mocap dataset [0]. Please refer to the shared response for details and results.
>
> **Further Comparisons**
>
> Thank you for the suggestions, we will extend our tables with additional results of related papers that were previously benchmarked on the Blender dataset, such as Nerfies.
>
> **TiNeuVox Features**
>
> The TiNeuVox features are used as an initialization and then further optimized during the learning process. We added additional ablations in the rebuttal PDF showing that, indeed, fine-tuning of TiNeuVox feature points is only necessary if they are randomly initialized. We optimize the color regressor ($\Phi_c$), density regressor ($\Phi_d$), pose regressor ($\Phi_r$), as well as the point features ($\mathbf{f}_i$), the point raw blend skinning vectors ($\mathbf{\hat{w}}_i$), and global scaling parameter $\alpha$. We will clarify this in the manuscript.
>
> **References**
>
> [0] Peng, Sida, et al. "Neural body: Implicit neural representations with structured latent codes for novel view synthesis of dynamic humans." Proceedings of the IEEE/CVF CVPR 2021.

---

> > ### Comment · Reviewer_u7Dh · 2023-08-20
> > **Thanks**
> >
> > Thank you for the answers to my question. I appreciate the additional experiments regarding initialization. It is interesting to see that fine tuning of features is not necessary at all, if they are initialized from TiNeuVox. I would have expected that the impact on quality is larger (in a negative sense). It seems to show though, that the feature spaces that PointNeRF and TiNeuVox come up with are quite compatible.
> >
> > I am not fully satisfied with the responses regarding additional comparisons. I welcome the qualitative results on ZJU-mocap but a quantitative comparison against previous human focused works would still improve the paper, even if the presented method performs worse. In the end, it is to be expected that the presented method will be slightly worse since it solves a harder task (without any human templates involved). It would be interesting though to see how large the gap is. Looking at the qualitative results it seems that the presented method has trouble to fine tune to the real data, leading to smoothing and some artifacts. I agree that it probably is due to issues in the data, e.g. incorrect camera poses. However, this makes a comparison even more important to assess the method.
> >
> > Regarding Nerfies and NSFF: can you confirm that the same conclusions still hold when you add them to the tables?
> >
> > All in all, my opinion has not changed in either direction after rebuttal and I tend to keep my original score.

---

> > > ### Author Response · Authors · 2023-08-21
> > >
> > > Thank you for your continuing support. We agree that the initialization experiment is interesting and that it shows that the features and their decoding are robust to the choice of their representation.
> > >
> > > We plan to expand our performance comparison with additional time-variant methods. To this goal, we were not able to find author’s benchmarks of NFSS and Nerfies applied to object-centric datasets that are the aim of our method. We could attempt to test these for the camera-ready revision if requested but we do not anticipate them to deviate far from performance of D-Nerf from the same era. Moreover, even a potentially higher reconstruction accuracy would not change our conclusions. This is because NFSS and Nerfies rely on unstructured backward deformation field or hyper-parametric canonical space, both of which are incompatible with the reposing goal. Hence, while the lack of kinematic constraints may allow for a better overfit to data, a good performance of these methods does not challenge our core contribution but instead hints at potential backbone alternatives.
> > >
> > > Instead of NFSS and Nerfies, we add additional comparisons to more recent methods Hex-Plane [2] and and K-Plane [3]. Table 1 shows that the reconstruction quality of these methods in the Blender dataset is in a similar range as TiNeuVox [1] shown in our submission. Therefore, the conclusions made in our manuscript remain valid.
> > >
> > > **Table 1: Average Performance on *Blender* with More Baselines**
> > >
> > >  |                      | PSNR $\uparrow$ | SSIM $\uparrow$ | LPIPS $\downarrow$ |      Reposeable       |
> > > |----------------------|-----------------|-----------------|--------------------|-------------|
> > > | D-NeRF [0]           | 30.50           | 0.95            | 0.07               | $\times$    |
> > > | TiNeuVox-B [1]       | 32.67           | 0.97            | 0.04               | $\times$    |
> > > | HexPlane [2]       | 31.04           | 0.97            | 0.04               | $\times$    |
> > > | K-Planes hybrid [3]  | 31.61           | 0.97            | -                  | $\times$    |
> > > | WIM [4]              | 23.81           | 0.91            | 0.10               | $\checkmark$|
> > > | Ours                 | 29.10           | 0.94            | 0.06               | $\checkmark$|
> > >
> > >
> > >
> > > **References**
> > >
> > > [0] Pumarola, Albert, et al. "D-nerf: Neural radiance fields for dynamic scenes." Proceedings of the IEEE/CVF Conference on Computer Vision and Pattern Recognition. 2021.
> > >
> > > [1] Fang, Jiemin, et al. "Fast dynamic radiance fields with time-aware neural voxels." SIGGRAPH Asia 2022 Conference Papers. 2022.
> > >
> > > [2] Cao, Ang, and Justin Johnson. "Hexplane: A fast representation for dynamic scenes." Proceedings of the IEEE/CVF Conference on Computer Vision and Pattern Recognition. 2023.
> > >
> > > [3] Fridovich-Keil, Sara, et al. "K-planes: Explicit radiance fields in space, time, and appearance." Proceedings of the IEEE/CVF Conference on Computer Vision and Pattern Recognition. 2023.
> > >
> > > [4] Noguchi, Atsuhiro, et al. "Watch it move: Unsupervised discovery of 3D joints for re-posing of articulated objects." Proceedings of the IEEE/CVF Conference on Computer Vision and Pattern Recognition. 2022.[5] Shao, Ruizhi, et al. "Tensor4d: Efficient neural 4d decomposition for high-fidelity dynamic reconstruction and rendering." Proceedings of the IEEE/CVF Conference on Computer Vision and Pattern Recognition. 2023.

---

### Official Review · Reviewer_RMHs · 2023-06-29

**Soundness:** 3 good
**Presentation:** 2 fair
**Contribution:** 2 fair
**Rating:** 6
**Confidence:** 4

**Summary:**

Authors presents a method to learn articulated model from multi-view video. And demonstrate an ability for efficient learning skeleton pose along as view synthesis model for dynamical structures.  Moreover the suggested method drastically improve convergence compared to naive approaches. To extract the skeleton authors apply Medial Axis Transform for initalization and RBF weights for the joints.
 The results on the D-NeRF dataset are demonstrate the superiority of this model over predecessor.

**Strengths:**

I appreciate the ideas in the method especially skeleton construction and mask loss.
- The approach doesn’t take into account pre-defined joint structure
- Convergence is quite fast
- Can handle non-rigid motions (in theory)
- This a first method learning articulation model purely from the input images
The description of the method is clear and solid.

**Weaknesses:**

The main idea is interesting, but the comparison is poor.
The idea of skelton initializing is mostly based on the previous works, and the it is not clear how much the final quality degrades, it is necessary to compare it with the ideal human skeleton, for example.
To apply this technique you first have to pre-train a dynamic version of NeRF, that complicates method compared to others.
All experiments are done for a single backbone and extremely limited variety of data
As you mentioned this method assume single object in the centre of the scene.
Simplified model produces unrealistic results (figure 6) the skeleton even cannot cover actions from training.
There are some works (e.g. TAVA) that incorporate idea of non-template methods (fairly they utilize keypoints) that have to be mentioned
Human related baselines can be interesting here.

**Questions:**

Why do you need  time-dependent rotations?
Does your training time include pre-training of the dynamic scene in Tab 2.
Do you need ARAP and transformation loss, how do they influence on the training.
Have you consider using something different to TiNeuVox method, there are plenty of much faster and accurate versions (e.g. HexPlane, Tensor4D)?

**Limitations:**

Highly relying on the retrained method
Speed and memory consumption for higher resolution
Simplification and skeleton construction itself is based on heuristic and can be very poor on some scenes.

---

> ### Author Rebuttal · Authors · 2023-08-09
>
> Thank you very much for your feedback! We answer your specific questions in this document. Please refer to the shared response for the experiment results and for answers to common questions.
>
> **Pre-trained Dynamic NeRF and Single Backbone**
>
> We have added additional results showing that our method can work with arbitrary backbones, if they yield a point cloud. Therefore, an initialization could also be achieved by a static pre-trained NeRFs or other means as in [0]. However, in our use-case we chose TiNeuVox because it can handle settings where a single view at a given time step is available which is the case for the Blender data set.
>
> **Comparison to TAVA**
>
> Thank you for pointing out TAVA [1]. We will discuss it as an additional example of an articulated reconstruction method. While TAVA utilizes keypoints instead of a full template, unlike our method, it still needs a tracked skeleton as an additional input. In our experiments we focus on comparisons to Watch-It-Move as the closest baseline because, like us, it does not need any prior skeletal information.
>
> **Need for Time-Dependent Rotations**
>
> Our method applies an Analysis-by-Synthesis approach. Therefore, we must learn the time-dependent rotations to warp the canonical representation to a time-specific representation to reconstruct the training images. Unlike TAVA we do not have access to tracked skeletons.
>
> **Does your Training Time Include Pre-Training of the Dynamic Scene?**
>
> Yes, all reported times do include the pre-training which is visualized by the horizontal offset of starting points for the blue “Ours” curves in Figure 4.
>
> **ARAP and Transformation Loss**
>
> ARAP enforces per-part rigidity by avoiding excessive weight mixing (please refer to Figure 8a in the main paper). Transformation loss enforces sparsity of joint rotations which helps with pruning in the post-processing step (please refer to Section 5 in the supplements)
>
> **Alternative backbones**
>
> Yes, please refer to the additional results where we show that random point initialization suffices for our method, therefore, Hexplane and Tensor4D could be utilized, too.
>
> **References**
>
> [0]: Xu, Qiangeng, et al. "Point-nerf: Point-based neural radiance fields." Proceedings of the IEEE/CVF CVPR 2022.
>
> [1] Li, Ruilong, et al. "Tava: Template-free animatable volumetric actors." ECCV 2022.

---

> > ### Comment · Reviewer_RMHs · 2023-08-16
> >
> > Thank you for the response! Since my score is lower than others, I will clarify my position after rebuttal.
> > I appreciate the comparison on the ZJU-Mocap and ask to add it into the main text as well as different backbones to improve the quality of the work for the readers.
> >
> > I am still interested in the initialization of the ideal skeleton for humans and comparing the learned LBS with TAVA. Moreover, a lot of implications are made with Blender dataset and can be unrelated to the real data (as demonstrated in the example of ZJU-mocap).
> > I wouldn't be against the paper, since the idea is interesting and the experiment design satisfies me, although, I would recommend authors to make their conclusions more fair by adding human data into the main text with comparison over relevant baselines.

---

> > > ### Author Response · Authors · 2023-08-18
> > >
> > > Thank you for your feedback. We will add an experiment to our manuscript, where we explore our method initialized with an ideal human skeleton similar to the one used in TAVA. Furthermore, we will incorporate the ZJU-Mocap experiments into the camera-ready manuscript. We hope that will allow you to adjust the recommendation in the official review form.

---

### Official Review · Reviewer_bAZS · 2023-07-07

**Soundness:** 3 good
**Presentation:** 3 good
**Contribution:** 3 good
**Rating:** 6
**Confidence:** 3

**Summary:**

The method reconstructs a reposable Dynamic NeRF of an articulated object from multiview videos. This is achieved by using linear blend skinning (LBS) of an automatically extracted skeleton to represent the deformation from canonical to observation space.

**Strengths:**

Combining LBS kinematics with NeRF appearance model effectively produces compelling reposing results.
Yielding a posable skeleton enables numerous traditional animation workflows.

**Weaknesses:**

A quick literature search yielded https://arxiv.org/abs/2208.14851 which talks about reposing with a different way of avoiding inverse mapping issues. It may warrant a mention in related work. Could that approach be combined with this paper's automatic skeleton extraction (they use a known rigged template mesh) to yield similar results?

Given that the method yields a posable skeleton, the results could have been much more compelling with artist-made or even mocap-driven animations (rather than just blending between a couple of poses).

**Questions:**

Thank you for the supplemental video. I think the Ground Truth T-Rex needs to also be animated.

Line 95 mentions Fi_d as a backward-warping MLP, did you mean Fi_b?

Line 98 uses p' in an expression for v_m, was that supposed to be p^c?

Specific descriptions of various MLPs listed in line 101 would be helpful.

In line 103, did you mean to use delta_i instead of just delta?

Is the N in equation 8 (L_arap) the same N as in L_smooth?

How come the simplified skeletons for the man and dinosaur are so sparse, compared to the two robots?

**Limitations:**

The authors address the main limitations of their approach, namely that LBS restricts the method to individual rigidly-linked bodies (as opposed to deformable objects or full scenes).

---

> ### Author Rebuttal · Authors · 2023-08-09
>
> Thank you very much for your feedback! We answer your specific questions in this document. Please refer to the shared response for the experiment results and for answers to common questions.
>
> **Relation to Dual-Space NeRF**
>
> Thank you for the suggestion, we will discuss Dual-Space NeRF [0] as an additional example of a template-based method for human bodies as it relies on the SMPL model for reposing. We note that our method differently targets general template-free reposing.
>
> **Reposable Skeleton Animation**
>
> We agree that making our pose representation more artist-friendly for example by directly applying motion captured data is an interesting open problem. We believe that future work in this direction will benefit from our contribution and build additional interfaces on top of it to enable intuitive manipulation of general articulated objects.
>
> **Reposed T-Rex**
>
> We have added an example showing a user-driven opening of the T-Rex’s mouth. Please refer to the PDF attachment Figure 2.
>
> **Specific descriptions of various MLPs**
>
> Thank you for the suggestion, we will add a table specifying the different MLPs in the appendix. We will also make our code publicly available upon acceptance.
>
> **Sparsity of Skeletons**
>
> Our Post-processing step prunes skeleton bones that are not needed to represent motion observed in the input video sequence. Therefore, for sequences with less motion, the skeleton may be reduced to only a few bones. This is the case for the T-Rex scene, for example, which only tilts the full body upward and opens its mouth.  We note that this is not a limitation but an intentional design feature that reduces the number of degrees of freedom for a potential animator, and that this step can be omitted if desired.
>
> **Other comments**
>
> We will correct the issues in lines 95, 98, and 103. The N in Eq. 8 and in $\mathcal{L}_\textrm{smooth}$ are indeed identical even though different neighborhood sizes could be considered if required.
>
> **References**
>
> [0] Zhi, Yihao, et al. "Dual-space nerf: Learning animatable avatars and scene lighting in separate spaces." 2022 International Conference on 3D Vision (3DV). IEEE, 2022.

---

> > ### Comment · Reviewer_bAZS · 2023-08-17
> > **Thanks for the rebuttal**
> >
> > My rating remains.
> >
> > A comment regarding "Reposable Skeleton Animation":
> >   You already have the capability of specifying new skeleton poses and rendering a 3D model from them. (quoting from line 231: "smoothly interpolating between user-defined poses").
> >    I was only suggesting to use an animation artist to specify more interesting poses to blend between.
> >    I don't think that's an open problem.

---

> > > ### Author Response · Authors · 2023-08-18
> > >
> > > Thank you for your feedback. We will add a more complex manually defined animation of a walking “Spot” robot to the manuscript to better showcase abilities of our method.

---

### Official Review · Reviewer_ZYjY · 2023-07-07

**Soundness:** 3 good
**Presentation:** 3 good
**Contribution:** 3 good
**Rating:** 4
**Confidence:** 5

**Summary:**

The paper presents an approach to articulated view synthesis, introducing the concept of Template-free Articulated Neural Point Clouds. The authors utilize a structure-free point-based NeRF representation which supports forward-warping of canonical objects to any poses through Linear Blend Skinning (LBS). Such a representation allows for the joint optimization of LBS pose parameters as well as dynamic NeRFs in a short training time. The method is evaluated on two datasets and compared with existing methods, such as D-NeRF, TiNeuVox, and WIM, showing superior or comparable performance on dynamic novel view synthesis and novel pose synthesis.

**Strengths:**

(+) The paper provides thorough explanations of the model and experiments.

(+) The paper proposed an effective point-based NeRF representation to support the forward-warping of the canonical space for template-free objects.

(+) The method achieves better novel view synthesis and reposing results in a shorter training time on two datasets compared to existing approaches. The visualized learned LBS weights fields and poses are cleaner than WIM.

**Weaknesses:**

(-) Inconsistent Motivation: The paper's motivation appears to be inconsistent throughout the text. Initially, the primary motivations are identified as 1) **Reposability** and 2) **Efficiency**. However, in the model section, the method is built upon a pre-trained TiNeuVox as initialization, aiming to animate TiNeuVox to new poses. Consequently, the efficiency of the proposed method heavily depends on the backbone/representation used, and the main motivation here seems to be **Reposability**. In the experimental section, the method primarily compares with WIM to demonstrate its efficiency over WIM on novel view/pose synthesis. A more convincing demonstration of the **Reposability** performance of the proposed method would involve comparing it with WIM in __its original training iterations__. To prove the **Efficiency** of the method, it would be beneficial to investigate the influence of different backbones/representations on the proposed method.

(-) Performance and Efficiency: The performance and efficiency of the proposed method do not seem promising compared to TiNeuVox. Given that the method builds upon a pre-trained TiNeuVox as initialization and aims to animate TiNeuVox to new poses, one would expect the method to perform at least similarly to TiNeuVox on the novel view synthesis task. However, as shown in Table 1 of the supplementary material, the method performs worse than TiNeuVox on 5 out of 7 scenes and requires significantly longer training time.

(-) Dependence on Pre-trained Model: The proposed method relies on a pre-trained model as initialization, whereas WIM does not require such a pre-trained model. For a fair comparison, it would be advisable to also compare with a modified version of WIM that is initialized from a pre-trained model. This comparison would provide a more balanced view of the strengths and weaknesses of the proposed method.

**Questions:**

* Please refer to the weaknesses section.

* How to optimize blend skinning weight vector $w_i$? And what is the motivation for defining another $\alpha$ for scaling the weights?

* $\Phi_d$ should be $\Phi_b$ in line 95 to represent the backward-warping MLP.

**Limitations:**

* The authors could discuss how variations in image quality, such as resolution, lighting conditions, and the presence of noise, might impact the performance of their method.

* How well does the method perform when applied to longer sequences or more complex scenes?

---

> ### Author Rebuttal · Authors · 2023-08-09
>
> Thank you very much for your feedback! We answer your specific questions in this document. Please refer to the shared response for the experiment results and for answers to common questions.
>
> **Inconsistent Motivation & Comparison to WIM with its Original Training Iterations**
>
> We added additional experiments showing that our methods can work with different backbones. Please refer to the shared response.
> As described in our paper, we modified the original scheduler of the WIM to access the full training data after the initialization phase. This was required because the original scheduler would access the full training sequence for the initial 80K iterations (around 11 hours with our GPU). This would invalidate the training curves in Figure 4.
>
> To demonstrate that this does not degrade WIM’s performance, we completed the full original training schedule with 200K iterations (~28h) for the robot ‘Spot’ using our data split. While it did eventually achieve a higher PSNR of 27.9 dB it did not get anywhere close to our 32.4 dB. Furthermore, it did this with almost 3x training time compared to our own method. The same holds for the second “merge” phase of WIM (a total of 350K iterations, ~ 2 days) which results in PSNR of 28.2 dB.
>
> Importantly, the PSNR of 25.9 dB achieved by WIM with its original scheduler after 80K steps (~11h) is comparable to that of our modified scheduler (26.3 dB) after the same number of iterations. This demonstrates that our comparison setup is fair.
>
> Please note that full training of WIM for multiple scenes is not possible within the rebuttal period which further highlights the difference in efficiency. Moreover, we were forced to reduce the batch size from 16 to 8 frames to fit into the 24 GiB VRAM of our RTX 3090 GPU.
>
> **Performance and Efficiency**
>
> Unlike TiNeuVox and other general dynamic NeRFs, our representation follows the Linear Blend Skinning which reduces the dimensionality of the pose space to enable reposing. This inevitably restricts the freedom for fitting the residual deformations that do not follow the low-dimensional model. We believe that explains the gap between the reposable and the non-reposable methods in Table 1 of the paper Supplement.
>
> Furthermore, the canonical volume in TiNeuVox is time-conditioned and, hence, it can lead to temporal changes not described by the deformation field alone [0]. This has been reported to further reduce the fitting error [1] but it goes against our idea of disentanglement between the canonical representation and reposing.
>
> We will make this distinction clear and highlight that our method achieves consistently better performance than the other reposable baseline, WIM. We believe that future research based on our method will bridge the gap by explicitly modeling the additional scene changes as pose-conditioned residuals.
>
> **Dependence on Pre-trained Models**
>
> While we agree that it would be interesting to study a similar initialization for other methods including Watch-It-Move, we note that the design of such an experiment is not obvious. The authors do not propose any way to initialize the representation based on a known density distribution and a trivial modification is not possible because, unlike our method, the Watch-It-Move representation is implicit. Moreover, it features an inherent ambiguity between the per-part ellipsoids $SDF_i(.)$ and the residual SDF MLP $S_\Theta(.)$. We argue that such a study is outside of the scope of the paper as we primarily focus on learning reposability.
> To demonstrate that our method is not closely tied to TiNeuVox, we conducted an experiment where we only initialize positions of the feature points but leave all feature values and network parameters random like in [3]. This partially approximates the behavior of WIM which spends the first 10K iterations to recover a coarse static representation from the first few sequence frames. Please refer to the shared response and the attached PDF.
>
> **Blend Skinning Vector and Alpha Scaling**
>
> The blend skinning vectors $w_i$ are optimized jointly end-to-end. The $\alpha$ parameter maps linear point-to-bone distances to a non-linear space, similarly as in [4]. Because $\alpha$ is global and shared by all blend skinning weights for all bones, it distributes gradient updates across the LBS model and reduces within-part variation of the effective skinning weights.
>
> **Impact of Dataset Image Quality, Length and Complexity**
>
> We conducted an additional experiment with the ZJU-Mocap dataset [5]. This is a challenging dataset because it contains longer and more realistic motion sequences and also realistic captured appearance, noise and complex lighting. Please refer to the shared response for details and results.
>
> **Additional comments**
>
> We will correct the typo in line 95.
>
> **References**
>
> [0]: Tretschk, Edith, et al. "State of the Art in Dense Monocular Non‐Rigid 3D Reconstruction." Computer Graphics Forum. Vol. 42. No. 2. 2023.
>
> [1]: Liu, Yu-Lun, et al. "Robust dynamic radiance fields." Proceedings of the IEEE/CVF Conference on Computer Vision and Pattern Recognition. 2023.
>
> [3]: Xu, Qiangeng, et al. "Point-nerf: Point-based neural radiance fields." Proceedings of the IEEE/CVF CVPR 2022.
>
> [4]: Yang, Gengshan, et al. "Viser: Video-specific surface embeddings for articulated 3d shape reconstruction." Advances in Neural Information Processing Systems 34 (2021): 19326-19338.
>
> [5] Peng, Sida, et al. "Neural body: Implicit neural representations with structured latent codes for novel view synthesis of dynamic humans." Proceedings of the IEEE/CVF CVPR 2021.
>
> [6] Li, Ruilong, et al. "Tava: Template-free animatable volumetric actors." ECCV 2022.

---

> > ### Comment · Reviewer_ZYjY · 2023-08-18
> >
> > Dear Authors,
> >
> > Thank you for your detailed rebuttal and the additional experiments provided. Your response has addressed several primary questions and concerns raised during the initial review.
> >
> > I do not hold a strong opposition to the acceptance of this paper given the merits of the ideas and the further clarified experimental comparisons provided in the rebuttal. I recommend incorporating the clarifications and improvements from the rebuttal into the revision.
> >
> > Best,
> >
> > Reviewer ZYjY

---

> > > ### Author Response · Authors · 2023-08-18
> > >
> > > Thank you for your comment and acknowledgment that important concerns have been addressed in the rebuttal. We hope that this will allow you to adjust the recommendation in the official review form. As suggested, we will add the additional experiments and results into the manuscript wherever the page limit allows.

---

### Official Review · Reviewer_KBgL · 2023-07-09

**Soundness:** 3 good
**Presentation:** 3 good
**Contribution:** 3 good
**Rating:** 7
**Confidence:** 5

**Summary:**

This paper tackles the task of dynamic novel view synthesis from multiview videos and aims for the ability of reposing. It tackles the problem with a point-based rendering approach. More importantly, it does not need any pre-defined or class-specific template/skeletons and learns a per-video data-driven skeleton. The effectiveness is verified on two datasets, i.e., Robots and Blender.

**Strengths:**

1. The paper is well-written and easy to follow.

2. The proposed skeleton learning is quite interesting: using a data-driven point cloud from TiNeuVox; then approximating an initial skeleton via Medial Axis Transform (MAT); finally using another data-driven component to refine the skeleton from MAT.

**Weaknesses:**

### 1. Robustness to Initialization

a. The proposed approach heavily relies on the NeRF pretraining: 1) geometry-wise, the initial skeleton needs the pre-trained NeRF's density function to sample points, which will be later used to run MAT; 2) appearance-wise, the points' features come from the pre-trained NeRF.

b. A natural question is how robust the proposed approach is to the initialization. One way to demonstrate this could be: 1) train TiNeuVox until convergence; 2) initialize the proposed approach from different checkpoints of TiNeuVox's training; 3) draw a plot of the final performance wrt different initializations.

c. What is related is: can authors clarify what the statement "we uniformly sample the density $\sigma$" (L124) means? Does it mean that you will first compute densities on plenty of points to get a sense of min and max densities? And how will you sample the point cloud with such min/max values?

### 2. Evaluation of Human Dataset

Arguably, one of the most important articulated view synthesis scenarios is human-related rendering. I am wondering whether authors can provide quantitative and qualitative results for the proposed approach on some human-related datasets, e.g., ZJU-MoCap used in [20]. Such results could provide a more complete assessment of the proposed approach.

**Questions:**

See above.

**Limitations:**

The authors provide a discussion on limitations.

---

> ### Author Rebuttal · Authors · 2023-08-09
>
> Thank you very much for your feedback! We answer your specific questions in this document. Please refer to the shared response for the experiment results and answers to common questions.
>
> **Robustness to Initialization**
>
> We conducted the proposed experiment, and we report the results in the shared response. They show that our method is robust to the quality of the initial geometry as long as functional parts are recovered.
>
> **Clarification Density Sampling**
>
> We will clarify that we sample the canonical density function $\Phi_d$ of TiNeuVox on a uniform coordinate grid and discard empty samples through thresholding. The grid resolution is adaptively chosen to retain approximately 10k points.
>
> **Human-Related Rendering**
>
> We tested our method with the ZJU-Mocap human capture dataset. Please refer to the shared section and the attached PDF for details.
>
> **Reliance on Pre-training**
>
> We included an additional experiment which shows that our method is robust to the choice of the backbone-specific initialization of the features and decoders. Please refer to the shared response "Impact of the Backbone Choice" for details.

---

> > ### Comment · Reviewer_KBgL · 2023-08-18
> > **Rebuttal Reply**
> >
> > I appreciate authors's effort in addressing my concerns.
> >
> > After reading the rebuttal and other reviews, I think the newly-added experiments provide a more complete evaluation of the proposed approach and make this a solid work. Therefore, I maintain my positive attitude toward acceptance.

---

### Author Rebuttal · Authors · 2023-08-09

We thank all reviewers for their valuable feedback, and we will use it to further improve our manuscript. We are glad that all reviewers found the description of our method clear, and that they appreciate that our method does not rely on any pre-defined skeleton (RMHs), offers better novel view synthesis with training time shorter than comparable methods (ZYjY), yields good and intuitive decomposition of object parts (u7Dh) and produces compelling reposing results (bAZS).

We made our best effort to answer the reviewer's questions and we conducted as many additional experiments analyzing the behavior of our method as possible within the limited time window of the rebuttal.

In this shared document we answer the most common points and present our additional experiments which are included in the attached PDF.

**Performance in challenging datasets and with human bodies**

We conducted an additional experiment with the ZJU-Mocap human body motion capture dataset [0]. This is a challenging dataset because it contains longer realistic motion sequences captured using cameras and human actors rather than synthetic renderings. Consequently, it also features a realistic appearance, capturing noise and complex lighting. We train our method on 5 sequences and we use 12 camera views for supervision. Each sequence consists of 490-790 frames, and we train for 320k iterations with a mask weight $w_1 = 0.2$.

The results are presented in the attached PDF in Figure 3. We observe that our method can recover the 3D shape as well as meaningful LBS model and pose animation. However, we do recognize that the image fidelity is perceptually lower than in the case of the synthetic datasets. We attribute this partially to the necessity to model a more complex appearance and partially to known inconsistencies in the dataset described by previous work (see the discussion of imperfect camera poses and inconsistent lighting in the Supplement F of [2]).

**Impact of the Backbone Choice**

While we used TiNeuVox as the backbone in all experiments in the paper, our method is designed to be agnostic to its design. In principle, any dynamic or even static 3D shape reconstruction method could be used to initialize our approach. While we can leverage pre-trained features and network weights obtained from the backbone, our method works well even if only a coarse feature-free point cloud is available. Therefore, we opted to test this as a more challenging and general question in favor of testing any specific backbone alternative.
To this goal, we follow the same training procedure for the Robots dataset as in the main paper but we only initialize positions of our feature points ($\mathbf{p}_i^c$) while keeping their feature values ($\mathbf{f}_i$) as well as the density ($\Phi_d$) and color regressors ($\Phi_c$) random before the start of our training.

We report quantitative results in the supplement PDF Table 1. We observe that the performance is comparable to the full initialization which suggests that any backbone that defines partitioning of the scene volume is an effective initialization for our method. Additionally, we found that increasing the weight of $\mathcal{L}_\textrm{skel}$ is beneficial for performance under these conditions. We will report these findings in the final version.

**Role of Feature Point Tuning**

In our method, we jointly train multiple components of our model in an end-to-end fashion. We experimentally assessed the contribution of fine-tuning the point feature values $\mathbf{f}_i$ in Table 2 of the supplement and we found that it does not have a major impact in the Blender dataset. However, it is important to note that this only holds because the feature values were initialized from an already trained feature decoding model of TiNeuVox. Fine-tuning features is necessary if they are initialized randomly (when using an alternative backbone), as demonstrated in the experiment above. Therefore, we keep the feature fine-tuning as a part of our method for generality, and we will clarify its role in the manuscript.

**Progressive Initialization**

While our previous experiments demonstrated that our method works well with a generic baseline, we conducted an additional experiment to assess how the quality and computational budget of the baseline training affect the performance of our downstream method. Specifically, we measure the effect of the TiNeuVox-backbone pre-train iterations on the initialization of our method in the Jumping-Jacks scene from the Blender data set.

We present the results in Figure 1 of the supplement. Surprisingly, after only 100 iterations of the backbone training, our method already achieves a PSNR score of over 30 dB.  This shows that even though such early initialization is very coarse, our method still recovers fine details. However, we acknowledge that our method would not be able to recover functional object parts completely missed during the skeleton extraction phase. Therefore, we opted to conservatively train the backbone for 20k iterations in all our results.

**References**

[0]: Peng, Sida, et al. "Neural body: Implicit neural representations with structured latent codes for novel view synthesis of dynamic humans." Proceedings of the IEEE/CVF CVPR 2021.

[1]: Noguchi, Atsuhiro, et al. "Watch it move: Unsupervised discovery of 3D joints for re-posing of articulated objects." Proceedings of the IEEE/CVF Conference on Computer Vision and Pattern Recognition. 2022.

[2] Li, Ruilong, et al. "Tava: Template-free animatable volumetric actors." ECCV 2022.

---

### Decision · Program_Chairs · 2023-09-21

**Decision:**

Accept (poster)

**Comment:**

This paper presents a data-driven method to discover an articulation model from a pre-trained dynamic nerf with the aid of efficient point-based rendering and LBS. It leads to It received mixed reviews. While most of reviewers found the idea of skeleton discovery interesting, there were concerned about 1) its generalization to different backbone dynamic nerfs other than TiNeuVox, 2) lack of results on real humans , 3) lack of comparisons with other D-nerfs. The authors presented additional results in the rebuttal to address these concerns. In particular, It reported experimental results on ZJU-Mocap with learned skinning weight visualization and compared with SOTA dynamic nerfs like Hexplane and K-plane. These results provided a more complete view of this work and helped addressed some of the raised concerns. As a result, the novel idea and solid design overweights the concerns. The AC would recommend its acceptance. Meanwhile, the AC urges the authors to incorporate the new results to the revision.